# Imitating Task and Motion Planning with Visuomotor Transformers

**Abstract:** Imitation learning is a powerful tool for training robot manipulation policies, allowing them to learn from expert demonstrations without manual programming or trial-and-error. However, common methods of data collection, such as human supervision, scale poorly, as they are time-consuming and labor-intensive. In contrast, Task and Motion Planning (TAMP) can *autonomously* generate *large-scale* datasets of *diverse* demonstrations. In this work, we show that the combination of large-scale datasets generated by TAMP supervisors and flexible Transformer models to fit them is a powerful paradigm for robot manipulation. We present a novel imitation learning system called OPTIMUS that trains large-scale visuomotor Transformer policies by imitating a TAMP agent. We conduct a thorough study of the design decisions required to imitate TAMP and demonstrate that OPTIMUS can solve a wide variety of challenging vision-based manipulation tasks with over 70 different objects, ranging from long-horizon pick-and-place tasks, to shelf and articulated object manipulation, achieving 70 to 80% success rates. Video results and code at https://optimustransformer.github.io/

**Keywords:** Imitation Learning, Task and Motion Planning, Transformers

## 1 Introduction

Large-scale data-driven learning, powered by the Transformer architecture [1], has transformed the fields of natural language processing (NLP) and computer vision (CV). Large models at the scale of billions of parameters, trained on massive corpi [2, 3, 4] exhibit powerful capabilities such as writing coherently [2, 5], answering questions [6], and image classification [7, 8] and generation [9]. Although there is recent work applying large Transformers to robot learning [10, 11, 12], the recipe of large-scale data-driven learning and Transformers has not yet achieved the same level of widespread success in robotic manipulation. One significant bottleneck is a lack of useful data – data collection is especially challenging because it requires the robot to interact in **real-time** with the world. Furthermore, not all data is useful: the collected interactions should be **relevant** for solving manipulation tasks of interest. Finally, for learned policies to be broadly applicable, they require access to a **diverse** set of task instances, which necessitates a **scalable** data collection pipeline.

Prior work has used human teleoperation [13, 14, 15, 16, 17, 18, 19] to collect large robot manipulation datasets, enabling training large scale models [20, 10]. However, this can require significant human time and labor – RT-1 [10] required 1.5 years of data collection. Other works have used reinforcement learning (RL) – this has the potential to scale more efficiently via autonomous data collection, but it is prohibitively expensive to run in terms of robot time due to its sample inefficiency [21, 22, 23, 24], and requires significant computation time and human reward engineering [25, 26]. In this work, we consider an alternative form of supervision, Task and Motion Planning (TAMP) [27], which addresses some key limitations of prior data-collection techniques. TAMP plans a discrete sequence of objects to interact with and how to manipulate them, and continuous motions that safely and correctly facilitate these interactions. TAMP supervision is beneficial because it: 1) collects data *autonomously* and 2) *efficiently* generates demonstrations by leveraging privileged

Submitted to the 7th Conference on Robot Learning (CoRL 2023). Do not distribute.

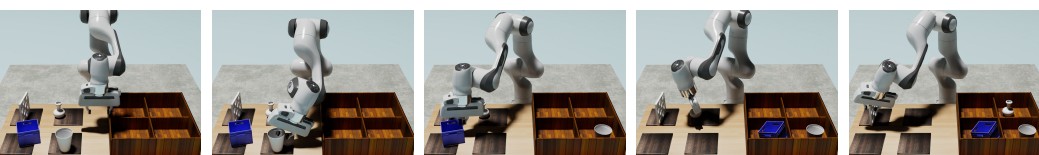

Figure 1: **Long-horizon task visualization.** We visualize the initial state and each intermediate pick state for the pick-and-place task. Note there is significant variation in geometry across each object, requiring the agent to perform a diverse series of grasps to complete the task.

information. TAMP can generate supervision on a wide distribution of task instances, producing **task relevant**, **diverse**, **large-scale** datasets for robot-learning.

However, TAMP on its own requires accurate estimation of the scene geometry and its state, is not reactive, and can spend significant time on planning. Instead, we propose to imitate TAMP across a wide range of tasks using closed-loop, visuomotor Transformer policies. As a result, we obtain **fast-to-execute**, **reactive** agents that can solve long horizon manipulation tasks **without state estimation**. Furthermore, by training on large, diverse datasets of successful trajectories, we show in our experimental evaluation that large Transformer policies have the capability of improving beyond TAMP performance. Finally, we note that while McDonald et al. [28] have also learned closed-loop policies from TAMP supervision, we perform an extensive study of the challenges in imitating TAMP, evaluate models across a wide range of tasks, and demonstrate novel capabilities including high-frequency end-to-end visuomotor control, task plan adaptation and scene generalization. Some challenges in imitating TAMP include learning from decisions made based on privileged information and multimodal demonstrations [29, 15].

To address these challenges, we propose **O**ffline **P**retrained **T**AMP **Im**itation **S**ystem, or OPTIMUS, a system for training visuomotor Transformer policies via imitation learning. **Our contributions are:**
- a novel framework for training visuomotor Transformer policies for high-frequency (30-50Hz) low-level control by taking advantage of TAMP supervision
- an empirically validated data-generation pipeline and study of the insights required to imitate TAMP
- strong results demonstrating that our trained policies can solve **over 300** long-horizon manipulation tasks involving up to **8 stages** and **72** different objects, achieving success rates of **over 70%**

## 2 Preliminaries

**Related Work:** OPTIMUS builds on a rich history of work in imitation learning and TAMP for robotic manipulation. In this work, we focus on the setting of offline learning via behavior cloning [30], in which a plethora of work has leveraged human demonstrations to learn effective policies [29, 31, 32, 33, 34, 35, 36, 37, 10, 20, 38, 39]. Our work instead relies on a TAMP supervisor, which can generate large, diverse datasets without human supervision. Furthermore, we build on recent work using Transformers for imitation [12, 40, 41, 10, 42, 43] by designing a fast to execute, visuomotor architecture operating over low-level control inputs. Finally, our system uses Task and Motion Planning [27, 44] to generate imitation data, a paradigm that has been recently explored in approaches that imitate planning [45, 46, 47] as well as TAMP directly [28]. In contrast to such prior work, our system adapts the TAMP data-generation process for improved imitation learning and uses a Transformer architecture that does not require any scene or task specific knowledge. See Appendix G for full related work.

**Background:** We address Partially Observable Markov Decision Processes (POMDP) $\langle \mathcal{S}, \mathcal{A}, \mathcal{T}, \mathcal{R}, p_0, \Omega, \mathcal{O}, \gamma \rangle$, where $\mathcal{S}$ is the set of environment states, $\mathcal{A}$ is the set of actions, $\mathcal{T}(s' \mid s, a)$ is the transition probability distribution, $\mathcal{R}(s, a)$ is the reward function, $p_0$ defines the distribution of the initial state $s_0 \sim p_0$, $\Omega$ is the set of observations, $\mathcal{O}(o \mid s)$ is the observation distribution, and $\gamma$ is the discount factor. We consider sparse reward POMDPs where $\mathcal{R}(s, a) \equiv -\mathbb{1}_{s \notin \mathcal{S}_*}$ is zero at terminal, goal states $\mathcal{S}_* \subseteq \mathcal{S}$ and elsewhere negative one. Solutions are *policies* $\pi_\theta(o_t, h_t)$ that operate on the history $h_t = (o_1, a_1, ..., o_{t-1}, a_{t-1})$ of observations $o \in \Omega$ and actions $a \in \mathcal{A}$, outputting the next action $a_t = \pi_\theta(o_t, h_t)$. The objective is to find a policy $\pi_\theta(o_t, h_t)$ that maximizes the expected

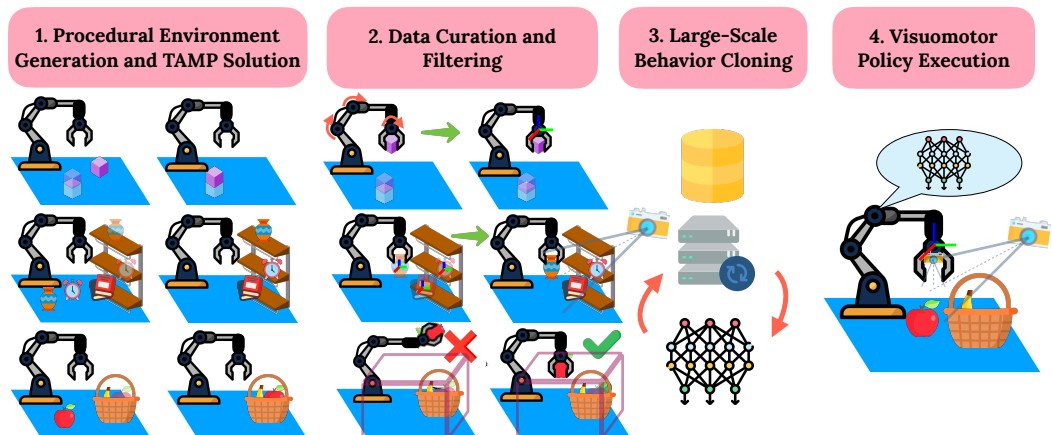

Figure 2: **OPTIMUS system**. *Column 1*: We generate a variety of tasks with differing initial configurations (*left*) and goals (*right*). *Column 2*: We transform TAMP joint space demonstrations to task space (*top*), go from privileged scene knowledge in TAMP to visual observations (*middle*) and prune TAMP demonstrations based on workspace constraints. *Columns 3 and 4*: We perform large-scale behavior cloning using a Transformer-based architecture and execute the visuomotor policies.

policy return $\mathbb{E}[\sum_{t=1}^{\infty} \mathcal{R}(s_t, a_t)]$. In this context, for behavior cloning, $\pi_\theta(o_t, h_t)$ is trained to regress $a_t$ from $(o_t, h_t)$ from a dataset $\mathcal{D}$ consisting of trajectories $\tau_i^n = (o_1^i, a_1^i, ..., o_{T_i}^i, a_{T_i}^i)$ produced by the expert, in which $i$ is the i-th trajectory in the dataset, $T_i$ is its length and $n$ is the n-th MDP.

**Task and Motion Planning:** TAMP algorithms address deterministic and observable, but hybrid, control problems [27]. In order to apply them to the POMDP for data collection, we grant them observability to the system state $s$. In simulation, this can be done through providing them access to the underlying simulator state. As a result, a TAMP policy $\pi_p(s_t)$ need only be a function of the state $s_t$, which is a sufficient statistic for the history $\langle h_t, o_t \rangle$. To construct this policy, we approximate the now observable POMDP with a deterministic model that can be effectively planned with [48]. Then, a TAMP algorithm uses this approximate model to plan a sequence of object interactions, the constraints present in each interaction (*e.g.* grasps and placements), and finally safe joint motions that realize them. An automated policy is built around the TAMP algorithm by tracking plans with a high-frequency feedback controller that outputs actions $a$ and periodically replanning [48].

Consider an example TAMP problem in which the goal is to place a **cup** on a **shelf** (*i.e.* the **Shelf** task). The TAMP model has the following parameterized actions: $\texttt{move}(q_1, \tau, q_2)$ moves the robot from configuration $q_1$ to configuration $q_2$ via trajectory $\tau$, $\texttt{pick}(o, g, p, q)$ picks object $o$ at placement pose $p$ with grasp pose $g$ when the robot is at configuration $q$, and $\texttt{place}(o, g, p, q, o_2)$ places object $o$ at placement pose $p$ on object $o_2$ with grasp pose $g$ when the robot is at configuration $q$. An example TAMP plan $p$ for the Shelf task is: $p = [\texttt{move}(\boldsymbol{q_0}, \tau_1, q_1), \texttt{pick}(\mathbf{cup}, g, \boldsymbol{p_0}, q_1), \texttt{move}(q_1, \tau_2, q_2), \texttt{place}(\mathbf{cup}, g, p, q_2, \mathbf{shelf})]$ The values in bold, the initial configuration $\boldsymbol{q_0}$ and cup placement $\boldsymbol{p_0}$, are constants. The other values are free parameters. A TAMP algorithm searches to find both the *plan skeleton*, the sequence of parameterized actions, as well as values for grasp $g$, placement $p$, configurations $q_1, q_2$, and trajectories $\tau_1, \tau_2$ that satisfy grasp, stability, kinematic, and collision constraints.

## 3 Designing a TAMP Imitation System

In this section, we motivate and describe our TAMP imitation system, OPTIMUS. We distill a privileged TAMP policy into a neural network in order to obtain policies that do not require access to state information, are fast to execute, and react instantaneously. To design OPTIMUS, we apply a TAMP supervisor to a procedural problem generator to produce demonstrations across a diverse range of tasks. However, trajectories produced by TAMP are not necessarily straightforward for an agent to imitate, especially when the agent must learn without access to privileged state information.

113 Consequently, we carefully create a data curation pipeline and couple it with agent design decisions
114 that maximize its ability to learn from TAMP trajectories and solve challenging manipulation tasks.

115 We consider tasks with significant variation across objects, poses, and configurations. We design four
116 environments: 1) block stacking, 2) single and multi-step pick and place, 3) shelf pick and place, and
117 4) articulated object manipulation with microwaves. To obtain object diversity, we load objects from
118 the ShapeNet dataset [49]. With a TAMP supervisor and diverse task distribution in place, we now
119 describe the data collection pipeline and how we use it for policy learning.

## 3.1 Cost-Minimizing TAMP

121 We use the PDDLStream planning framework [50] to model the TAMP domain and the *adaptive*
122 algorithm, a sampling-based algorithm, to plan. Our formulation makes use of samplers for grasp
123 generation, placement sampling, inverse kinematics, and motion planning. The samplers can produce
124 a large, if not infinitely large, set of diverse values. We implement the grasp generator using the
125 ACRONYM grasp dataset [51] for ShapeNet objects. We use TRAC-IK [52] for inverse kinematics
126 (IK), and bidirectional Rapidly-Exploring Random Trees (BiRRT) [53] for motion planning.

127 When using TAMP solutions for imitation learning, it is essential to train on high-quality plan
128 traces. Behavior cloning techniques typically are adverse to multi-modal policy behavior, so a TAMP
129 demonstrator that takes several different actions at a particular state produces data is challenging to
130 imitate. One way to reduce TAMP policy variability is to optimize for low-cost plans. Although
131 a TAMP problem is not guaranteed to have a unique minimum cost solution, this strategy biases
132 solutions to a consistent family of low-cost plans.

133 We propose a two-stage approach to producing low-cost TAMP solutions. First, we use cost-sensitive
134 PDDLStream planning that minimizes the joint-space distance traveled. Specifically, we define
135 costs for $\texttt{move}(q_1, \tau, q_2)$ actions that limit $\infty$-norm (max) of the distance $||q_1 - q_2||_\infty$ between
136 configurations $q_1$, $q_2$. The straight-line distance between two configurations is a lower bound on
137 the length of the shortest collision-free path between them. We optimize this lower bound before
138 performing motion planning which is computationally expensive due to continuous collision checking.
139 This PDDLStream algorithm is asymptotically optimal [54, 50], but it might take arbitrary long to
140 find a plan below a target cost bound. In practice, we run the planner in an anytime mode with
141 a computation budget of five seconds and return the best plan identified. In the second stage, we
142 perform motion planning using BiRRT; however, it can produce motions that are jagged and locally
143 sub-optimal. To smooth these trajectories, we post-process them using cubic spline short cutting with
144 velocity and acceleration limits [55], which converges to a locally time-optimal trajectory.

145 Finally, we aim to limit the variability in IK solutions. This is also advantageous for task-space
146 control, which lacks the control authority to reach all IK solutions. We seed TRAC-IK's optimization-
147 based IK from a single configuration seed, the initial configuration, and optimize for the closest
148 solution to the initial configuration within a 10 millisecond timeout. This also biases TAMP toward
149 plans that stay near the initial configuration, typically accelerating the search for low-cost plans. By
150 intentionally not exploiting the redundancy to explore diverse IK solutions, we limit the completeness
151 of the TAMP algorithm for the benefit of downstream learning.

## 3.2 Generating Imitation Data from TAMP

153 Directly training on datasets collected by TAMP is a challenge for imitation learning, as the TAMP
154 system operates with access to information unavailable to the learner, controls the robot in joint
155 space, which can be difficult to learn in, and generates demonstrations that may not necessarily take
156 the shortest path in task-space. To address these issues, we highlight design decisions regarding the
157 observations and actions we produce from the TAMP data-generation process as well as how we
158 select which demonstrations to train on.

**Imitating a Privileged Expert:** TAMP operates over a privileged view of the world. It has access to
160 information that is difficult to obtain from a perception system, such as environment geometry and
161 object state. To address these issues, OPTIMUS operates over image observations by using multiple
162 camera views in each task (1-2 fixed cameras and 1 wrist-mounted camera). We find that multiple

views, in particular the wrist camera, help the agent to better perceive scene geometry [56] and align its actions with the privileged expert. By training over multi-view RGB observations, we provide the network with an observation space that is invariant to object symmetry, encodes 3D information, is efficient to train over, and enables simplicity of the architecture.

**Learning from TAMP Generated Actions:** The TAMP system plans arm motions in configuration space, in which it can fully control each robot degree of freedom. However, training vision-based policies in joint-space is difficult due to the challenge of learning the camera projection from pixels to poses and then the redundant inverse kinematics mapping from pixels to joint angles [57, 58, 29]. Additionally, for robots with more than six degrees of freedom, joint space is higher dimensional than task space. Thus, in OPTIMUS, we instead use task-space control. We generate task space trajectories by performing forward kinematics on joint-space way-points given by the TAMP planner, then execute an operational-space (task-space) controller [59] to achieve them. Appendix C conducts an experiment comparing the trained policy success rate with joint-space actions versus task-space actions. Fig. C.1 shows that task-space actions enable higher success rates.

**Filtering Demonstrations:** Since there is variance in run-time due to random sampling and the TAMP system is not guaranteed to converge in plan cost within the fixed time limit, some plans and thus behaviors may be sub-optimal. This data can often hamper policy learning by operating outside of the space of nominal solution trajectories. Training on this data from the TAMP system increases the likelihood of the agent leaving areas of high state space coverage, which produces policies that exhibit heightened compounding error. To ease the burden on the policy, we curate the data using several trajectory pruning rules. During data collection, we employ joint-space path smoothing. However, straight-line paths through joint-space are non-linear in task space, resulting in longer motions in the learner's action space. Therefore, we propose two data pruning rules (Fig. 2 *column 2*) to filter TAMP demonstrations. First, we remove outlier trajectories that have task-space length greater than two standard deviations away from the mean trajectory length, which can be viewed as randomly restarting TAMP episodes to reduce plan variance. Second, we impose a containment constraint in the form of a bounding box in visible workspace and prune out trajectories in which the end-effector pose exits the box. Appendix C and Fig. C.1 illustrate that the combination of these rules does improve performance by comparing the trained policy success rate with and without filtering.

### 3.3 Training Imitation Policies at Scale

We now describe the imitation pipeline in OPTIMUS. Given large, diverse datasets from TAMP, we perform offline behavior cloning to distill the TAMP expert into a visuomotor policy.

**OPTIMUS Architecture:** Our policy must operate over a history of multiple camera views and proprioception, output low-level task space actions, and execute in real time. To that end, we design our policy $\pi$, visualized in Fig. 3, as a Transformer operating over a history of observations $h$, in which each token corresponds to a single observation time-step. As a result, the Transformer can efficiently attend to all observations as the Transformer context length is set to $h$. To produce a single input token for a time-step $t$, we first embed each input, images from cameras $1, ..., N$ ($I_1^t$ through $I_N^t$) as well as proprioception $p_t$, into fixed dimensional vector spaces. For proprioception, we pass in the end-effector pose (xyz position and quaternion orientation) and gripper joint position (dual finger positions), encoded by an MLP. For embedding images, we use the vision backbone from Mandlekar et al. [29]: ResNet-18 [60] with a spatial softmax [61] output activation. We then fuse the inputs for a single time-step to produce $z_t$, a vector matching the Transformer

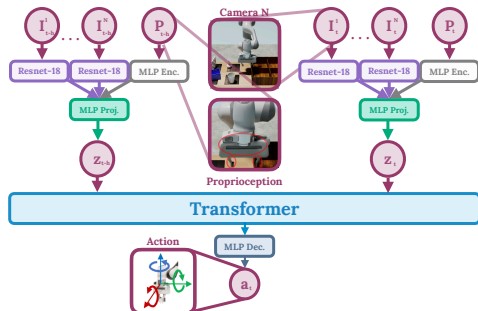

Figure 3: **OPTIMUS policy architecture**. The model takes as input multiple images and proprioception information per time-step, with a context of $h$. We encode the input using Resnet-18 for images and a MLP for the low-dimensional observations. We concatenate the embeddings, project them into the Transformer embedding dimension and pass them to the Transformer, which predicts an embedding that is decoded into an action.

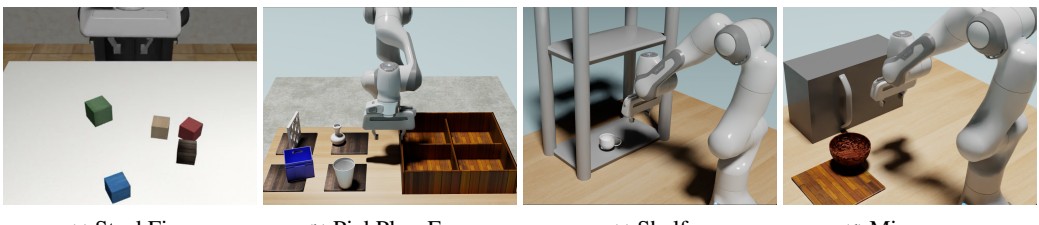

| (a) StackFive | (b) PickPlaceFour | (c) Shelf | (d) Microwave |

Figure 4: **Environment Visualizations**. We evaluate OPTIMUS on long-horizon block stacking (a), multi-step pick-place (b), shelf object manipulation (c), and articulated object manipulation (d).

embedding dimension, by concatenating and performing an MLP projection. The Transformer attends to each token $z_t$ and outputs a distribution over action $a_t$ corresponding to the current time-step.

The data distribution outputted by the TAMP supervisor is heavily multi-modal, from the diversity in planned paths to the variety of grasps and placements per object. As a result, we use a Gaussian Model Mixture (GMM) output distribution with $K = 5$ components for the policy from Mandlekar et al. [29] and train the model using log likelihood. As in [29], we find that this loss function provides a significant improvement over the standard MSE loss, which produces a unimodal policy.

## 4  Experimental Evaluation

In our experimental evaluation of OPTIMUS, we aim to answer the following questions: 1) Can imitating TAMP enable end-to-end policies to acquire long-horizon behaviors? 2) Does TAMP allow networks to solve complex manipulation tasks involving 3D obstacles and articulated objects? 3) Does diverse environment generation along with TAMP data-collection enable large-scale behavior learning? We begin by describing the datasets, tasks, and protocols that we use to evaluate OPTIMUS. We then proceed to experimentally evaluate OPTIMUS.

**Datasets and Tasks:** We evaluate OPTIMUS across block stacking, pick and place, shelf manipulation and articulated object manipulation. (Fig. 4). Our block stacking tasks have two (Stack), three (StackThree), four (StackFour) or five (StackFive) blocks, with 1K, 2K, 5K, and 7K demonstrations respectively. For pick and place, we have: PickPlace-1, pick-place with a single object using 1K demos, and pick-place with two (PickPlaceTwo), three (PickPlaceThree), and four objects (PickPlaceFour) into separate bins. Finally we have two tasks in which the goal is for the agent to move the object to the target location while maneuvering in tight spaces (Shelf-1) or first pulling open a microwave door (Microwave-1), for which we generate 1K demonstrations each. For PickPlace, Shelf, and Microwave, we additionally evaluate two multi-task variants, in which we sample a set of 19 and 72 objects from ShapeNet. We collect a 1K demonstrations per object, with 19K and 72K total trajectories, resulting in the following datasets: Pickplace-19, PickPlace-72, Shelf-19, Shelf-72, Microwave-19, Microwave-72. See Appendix Sec. D for complete task descriptions and details.

**Evaluation Protocol:** We evaluate BC-MLP [62] and BC-RNN [29], which consist of a Resnet-18 backbone followed by an MLP and an LSTM [63]. Additionally, we compare against Behavior Transformer (BeT) [43], which discretizes the dataset into clusters using K-Means and uses a Transformer model to predict a cluster center and an offset, in order to handle multi-modal data. Each method uses on the order of magnitude of 30M parameters, uses the same architecture as OPTIMUS for the vision-backbone, and leverages the data-pipeline we propose in Sec. 3. For each task, we evaluate on a dataset of unseen initial environment states. For single-task results, we evaluate using 50 problems and average across 3 random seeds per run. For multi-task results, we evaluate using 10 problems per task, with a single seed per run. We use task success rate as our evaluation metric, which is 1 if all objects are in a goal arrangement and 0 otherwise.

### 4.1  Learning Results

We first show that OPTIMUS can imitate the TAMP system to high fidelity on simple, shorter horizon tasks. We then extend our evaluation to the long-horizon regime in which the task complexity grows significantly with the number of objects. Next, we move beyond the table-top manipulation setting

| Dataset | BC-MLP | BC-RNN | BeT | OPTIMUS |
|---|---|---|---|---|
| Stack | 100 | 100 | 100 | 100 |
| StackThree | 98 | 88 | 76 | **100** |
| StackFour | 83 | 77 | 61 | **96** |
| StackFive | 57 | 57 | 45 | **70** |

| Dataset | BC-MLP | BC-RNN | BeT | OPTIMUS |
|---|---|---|---|---|
| PickPlaceTwo | 96 | **98** | 80 | 96 |
| PickPlaceThree | 62 | 81 | 46 | **91** |
| PickPlaceFour | 33 | 38 | 22 | **60** |

Figure 5: **Long Horizon Manipulation Results.** (left) Performance is shown in terms of task success rate. While all methods are able to solve single-step block stacking, only OPTIMUS is able to solve longer-horizon variants. (right) For long-horizon manipulation, while the baselines are competitive with OPTIMUS on PickPlaceTwo, OPTIMUS demonstrates significant improvement in success rate as the number of objects increases.

and train policies to solve tasks involving a shelf and microwave. Finally, we demonstrate that OPTIMUS can enable multi-task policies that can manipulate a wide range of objects. Please see Appendix C for a detailed analysis and ablation of OPTIMUS.

**OPTIMUS imitates TAMP to high fidelity on simple pick-and-place tasks.** On Stack (Fig. 5), we find that OPTIMUS and the baselines are all able to achieve 100% performance on the task. On the other hand, on PickPlace-1, (Fig. 5), while the baselines achieve high success rates of up to 97%, only our method is able to solve the task at 100% success rate. These results demonstrate that on simple tasks, OPTIMUS can fit well to the output of the TAMP system, even though OPTIMUS does not have access to any privileged information. We note that even with significant tuning, BeT struggles to fit to TAMP data on most of our tasks. We hypothesize that this may be due to the difficulty of fitting K-Means as the dataset size increases, especially as TAMP generated datasets contain on the order of 1-100K trajectories depending on the task. As a result, the cluster centers can be highly inaccurate, increasing the burden on the transformer to fit appropriate offsets.

**OPTIMUS enables visuomotor policies to solve manipulation tasks with up to 8 stages.** We first evaluate on long-horizon block stacking, a task that is difficult because the stack of blocks becomes more unstable as its height grows. We train visuomotor policies across StackThree, StackFour, and StackFive, and visualize the results in Fig. 5. OPTIMUS outperforms the baseline methods while achieving near-perfect performance across each task. Multi-step pick-place is even more difficult as the network must learn to fit a variety of different grasps for each object. We plot the results for the multi-step pickplace tasks in Fig. 5. We find that while BC-RNN outperforms OPTIMUS on PickPlaceTwo, OPTIMUS exhibits a large performance improvement on PickPlaceThree and Four. These results demonstrate that with either primitive or general-purpose rigid objects, it is possible to train policies to perform long-horizon behaviors consisting of up to 8 pick and place operations or 40 TAMP high-level actions, with high success rates of **70%** and **60%** respectively. An important take-away from these results is that for longer-horizon tasks, the Transformer policy architecture we develop in OPTIMUS greatly outperforms MLPs and RNNs.

| Guided TAMP | OPTIMUS |
|---|---|
| 88 | **90** |

Table 1: **Comparison against Guided TAMP.** Results are in terms of task success rate.

We additionally compare against prior work on imitating TAMP [28] on the Robosuite [58] PickPlace task, which involves picking and placing four fixed objects: a milk carton, a soda can, a cereal box and a piece of bread, into separate bins. In contrast to PickPlaceFour, Robosuite PickPlace can be solved with top-down, axis-aligned grasps due to the simplicity of the object geometry. However, the initial configurations are more challenging as all the objects are placed together in the same bin. We generate 25K demonstrations of the task using our system. As we show in Table 1, OPTIMUS achieves favorable results to Guided TAMP (90% vs. 88%) without requiring access to privileged state information, a fixed set of ground actions or online supervision.

**OPTIMUS can also solve tasks requiring obstacle awareness and skills beyond pick-and-place.** On Shelf-1, OPTIMUS is able to grasp then place the object in the middle rung of the shelf at high success rates of 91% shown in Table 2. On Microwave-1 OPTIMUS outperforms the baselines by nearly 10%, achieving 86% success rate overall. This is likely because OPTIMUS is able to better fit the data in the multi-step manipulation regime, as noted in the prior section. The results on the Shelf and Microwave tasks demonstrate that OPTIMUS can learn to solve difficult manipulation tasks that require obstacle awareness and the ability to manipulate articulated objects.

**OPTIMUS can learn to adapt its behavior based on the scene configuration.** As we describe in the Appendix, OPTIMUS is able to learn to adapt its task plan to produce additional stacking operations (StackAdapt) or clear the area in front of the microwave (MicrowaveAdapt) achieving 96% and 75% success. OPTIMUS is able to generalize to unseen receptacle sizes, achieving 80% and 70% success rate on held out shelves and microwaves.

We next evaluate the ability of our TAMP generation pipeline to collect diverse datasets in order to train large-scale policies. We add variety in the form of objects with differing geometries, requiring a single network to learn a range of manipulation behaviors end-to-end.

| Dataset | BC-MLP | BC-RNN | BeT | OPTIMUS |
|---|---|---|---|---|
| PickPlace-1 | 94 | 97 | 85 | **100** |
| PickPlace-19 | 61 | 58 | 50 | **85** |
| PickPlace-72 | 50 | 49 | 41 | **75** |
| Shelf-1 | **91** | 88 | 70 | **91** |
| Shelf-19 | 48 | 31 | 26 | **66** |
| Shelf-72 | 30 | 36 | 13 | **48** |
| Microwave-1 | 73 | 77 | 51 | **86** |
| Microwave-19 | 24 | 41 | 31 | **61** |
| Microwave-72 | 23 | 29 | 16 | **47** |

Table 2: **Single and Multitask Results across Pick-Place, Shelf, Microwave.** Performance is shown in terms of task success rate. While the baselines are competitive with OPTIMUS on the single task variants of each task, OPTIMUS greatly outperforms the baselines as the number of objects increases across all tasks.

**OPTIMUS achieves high success rates on vision-based manipulation tasks with up to 72 objects.** For each task: PickPlace, Shelf, and Microwave, we evaluate on their 19 and 72 object variants (Table 2). On the 19 object tasks, OPTIMUS achieves 85%, 66%, and 61% in greatly outperforming the best baseline for each task: 61%, 48%, and 41%. Similarly on the 72 object tasks, we find that OPTIMUS obtains 75%, 48% and 47% success rates, in comparison to 50%, 36% and 29% for the best baseline. From these results, we note two important points: 1) Transformer-based architectures such as OPTIMUS are highly effective for multi-task imitation learning: they greatly outperform MLPs and RNNs. 2) While the single task variants of these tasks are solved at high success rates, performance drops significantly in the multi-task case, particularly for more challenging manipulation tasks such as Shelf and Microwave, indicating further work remains to bridge that gap.

Finally, we highlight three advantages of OPTIMUS over the TAMP system: 1) success rate improvement over the TAMP supervisor, 2) faster run-time, 3) operation from image instead of state input. We evaluate TAMP and OPTIMUS on all of the single task datasets and find that on average, OPTIMUS almost doubles the performance of the TAMP supervisor (87% vs. 52%). Additionally, we evaluate the run-time of OPTIMUS against TAMP by computing the average time per step for both systems across 100 trials. Overall, OPTIMUS is 5-7.5x faster than TAMP (21/31ms vs. 150ms per action). See Appendix Sec. B for analysis of scene adaptation and TAMP comparison results.

# 5 Limitations and Future Work

In this work, we propose an approach for distilling privileged TAMP experts into large-scale visuomotor policies. We generate large, diverse datasets and train high-capacity Transformer models to solve challenging, long-horizon manipulation tasks without task, state, or environment knowledge. Even so, there are limitations to OPTIMUS and scope for future work. First, for OPTIMUS to be able to solve a task, TAMP needs to be capable of solving it at training time, which could prevent OPTIMUS from being applied to tasks that require considering dynamics or tasks involving contact-rich manipulation, which can be challenging for traditional TAMP. However, TAMP can be applied to such tasks, e.g. pouring, scooping, stirring, peg insertion, or coffee making, by leveraging integrated task planning and skill learning approaches [64, 65, 66, 67], which OPTIMUS can leverage for supervision as well. Second even with OPTIMUS, there is a significant drop in success rate with increasing task difficulty and number of objects (Sec. 4.1), which suggests further work on multi-task learning is necessary to bridge that gap. Finally, we describe two possible approaches to applying OPTIMUS to the real world: 1) Collect TAMP data in a controlled setting at training time using a pose estimation system (e.g. AR tags, SAM with calibrated depth [68], MegaPose [69]) and distill using OPTIMUS. At test time, OPTIMUS would not require pose estimation, while providing significantly faster execution and superior task performance. 2) Transfer imitation policies trained in simulation either zero-shot or via fine-tuning to real data [70, 71, 72, 73]. We leave this extension of OPTIMUS to future work.

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

# Appendix

## A    Table of Contents

- **Additional Learning Results** (Appendix B): Additional experimental results demonstrating OPTIMUS's effectiveness on more tasks and additional baselines.
- **Ablations** (Appendix C): Ablations and analyses of OPTIMUS, demonstrating the effectiveness of our design decisions.
- **Environments** (Appendix D): Description of all the environments we use in this work.
- **Agent Structure** (Appendix E): details regarding the observation space and action space of the agent.
- **Experiment Details** (Appendix F): Full details on how OPTIMUS is implemented, specifically the hyper-parameters used for training and network architectures.
- **Related Work** (Appendix G): Full description of the related work.

## B Additional Learning Results

**OPTIMUS exhibits multi-task category control capabilities.** We extend our multi-task results to the setting in which the task category can also vary by training a multi-task category model on a dataset of demonstrations from Pickplace, Shelf and Microwave. Across the tasks, the goal is implicitly communicated by the initial observation. In this setting, we use the same camera views across all tasks: the left/right shoulder views and the wrist camera. We build a dataset of 15K trajectories with 5 objects per task category and 1K demos per task. We include the results in Table B.1. Similar to our multitask results in the main text, we find that OPTIMUS is able to demonstrate multi-task category capabilities: a single Transformer is capable of learning to pick and place objects on a table, manipulate objects in a shelf, and open doors across a large set of objects at a success rate of 73%. This experiment shows that OPTIMUS greatly outperforms the baselines on multi-task learning.

| Dataset | BC-MLP | BC-RNN | BeT | OPTIMUS |
|---|---|---|---|---|
| PickPlace-Shelf-Microwave | 44 | 47 | 41 | **73** |

Table B.1: **Multi-task category results.**. By distilling TAMP demonstrations across three environments (PickPlace, Shelf, and Microwave), OPTIMUS is able to effectively manipulate a wide array of objects across diverse scenes, purely from image input.

**OPTIMUS can learn to adapt its behavior based on the scene configuration.** We evaluate OPTIMUS on two tasks that involve adapting the task plan based on the configuration of objects in the scene: StackAdapt and MicrowaveAdapt, and two that require adapting motions to randomized receptacle sizes: ShelfReceptacle and MicrowaveReceptacle. As shown in Table B.2, OPTIMUS is able to effectively leverage visual input to learn when additional stacking operations are needed (StackAdapt) or when the area in front of the microwave needs to be cleared (MicrowaveAdapt), achieving 96% and 75% respectively, compared to the best baseline (96% and 40%). Additionally, we demonstrate that OPTIMUS is able to effectively learn to generalize to unseen receptacle sizes with high success rates, achieves 80% and 70% on held out shelves and microwaves respectively. These results illustrate that OPTIMUS can distill scene conditioned task plan adaptation and motion generalization across scene configurations from TAMP supervision.

| Dataset | BC-MLP | BC-RNN | BeT | OPTIMUS |
|---|---|---|---|---|
| StackAdapt | **96** | 92 | 81 | **96** |
| MicrowaveAdapt | 25 | 40 | 13 | **75** |
| ShelfReceptacle | 72 | 71 | 59 | **80** |
| MicrowaveReceptacle | 48 | 55 | 31 | **70** |

Table B.2: **Scene-based adaptation results**. OPTIMUS can learn to vary the task plan it executes based on the scene configuration *(rows 1 and 2)* as well as adapt to unseen receptacles *(rows 3 and 4)*.

**OPTIMUS solves tasks that RL methods fail to make progress on.** We perform a thorough comparison of OPTIMUS against modern deep RL methods across four benchmark tasks in Robosuite (Stack, PickPlaceCan, PickPlaceCereal, PickPlace), for which there exist dense rewards suitable for RL. We evaluate 3 algorithms: SAC [74], a commonly used off-policy model free method, DRQ-v2 [75], a state-of-the-art vision-based RL method, and MoDem [76], an efficient visual model-based RL method. We train each RL method with up to 5 million samples with 5 seeds. We show the results in Table B.3. Across every task, the RL baselines struggle to learn the long-horizon behaviors, failing to achieve a greater than 10% success rate on any given task. These environments pose a significant exploration challenge for RL agents, especially when trying to map high-dimensional observations such as images to low-level control actions.

**OPTIMUS can outperform purely Transformer based architectures.** In this experiment, we integrate Transformer-in-Transformer [77], a recently proposed Transformer architecture for control, into OPTIMUS and evaluate it across five tasks: Stack, Pickplace-1, Shelf-1, Microwave-1 and

| Dataset | SAC | Drq-v2 | MoDem | OPTIMUS |
|---|---|---|---|---|
| Stack | 0 | 6 | 3 | **100** |
| PickPlaceCan | 0 | 10 | 0 | **100** |
| PickPlaceCereal | 0 | 5 | 0 | **100** |
| PickPlace | 0 | 0 | 0 | **90** |

Table B.3: **Comparison of OPTIMUS vs. RL methods**. OPTIMUS is able to solve each Robosuite task to a high success rate, while RL methods struggle to make progress due to exploration challenges.

PickPlaceFour. We do so by modifying OPTIMUS to use the code released by the authors of [77] as the Transformer block. The default settings from [77] did not perform well on our tasks (20% success rate on Stack), so we made the following modifications: We modify the backbone used in [77] by increasing the number of layers in the Vision Transformer backbone from 1 to 6, the number of heads from 1 to 4, the patch dimension from 84 to 19 (to obtain a 4x4 grid). With these settings we achieve 54% success rate on Stack. We then perform one further modification: instead of using the class token output as the state representation in [77], we reshape the tokens corresponding to each patch into 4x4 images and then pass them through a spatial softmax to obtain a keypoint representation of the image. Doing so improves the performance of Transformer-in-Transformer from 54% to 86% on the Stack task. We run Transformer-in-Transformer across all five tasks and include the results against OPTIMUS in Table B.4. Across each task, we find that OPTIMUS is able to outperform Transformer-in-Transformer, with an average performance improvement of 16.8%. One additional advantage of our architecture over the one proposed in [77] is that ours is 4-5x faster to execute. We hypothesize that a likely reason for this performance discrepancy is that on our visuomotor control tasks, ResNets [60] are a powerful inductive bias. They maintain spatial locality which allows the spatial softmax [61] to easily identify important key-points in the image.

| Dataset | Transformer-in-Transformer | OPTIMUS |
|---|---|---|
| Stack | 86 | **100** |
| PickPlace-1 | 82 | **100** |
| Shelf-1 | 73 | **91** |
| Microwave-1 | 67 | **86** |
| PickPlaceFour | 45 | **60** |

Table B.4: **Comparison of OPTIMUS vs. Transformer-in-Transformer**. OPTIMUS is able to outperform purely Transformer based architectures such as Transformer-in-Transformer [77] by 16.8% across 5 tasks, demonstrating that our architecture is well-suited to imitating TAMP data from visual input.

We describe and empirically validate three advantages of the distilled policies over the TAMP system: 1) success rate improvement over the TAMP supervisor, 2) faster run-time, 3) operation from perceptual instead of state input.

**OPTIMUS almost doubles the performance of the TAMP supervisor.** To evaluate TAMP, we execute 50 trials averaged over three random seeds on each single-task environment and record the performance in Table B.5. We find that OPTIMUS is able to outperform the TAMP system by a wide margin, from 20% on the easiest task, PickPlace, to 64% on Microwave-1 and 44% on the hardest task, PickPlaceFour. TAMP with joint space control has better performance on average than TAMP with task space control (52% vs. 45%), but still performs significantly worse than OPTIMUS (52% vs. 87%). We instead find that not all grasps execute perfectly every time, likely due to differences in simulation, planning and control schemes from the ACRONYM paper. As a result, we observe grasp execution failures and object slippage during placement motions. OPTIMUS avoids learning these failure cases by only distilling the successful trajectories, which enables it to successfully generalize to unseen configurations of the task.

**OPTIMUS executes 5-7.5x faster than TAMP.** We evaluate the run-time of OPTIMUS against TAMP by computing the average time per step for both systems across 100 trials. We run the

| Dataset | TAMP-joint | TAMP-task | OPTIMUS |
|---|---|---|---|
| PickPlace-1 | 82 | 82 | **100** |
| PickPlaceTwo | 52 | 58 | **96** |
| PickPlaceThree | 40 | 50 | **91** |
| PickPlaceFour | 34 | 16 | **60** |
| Shelf-1 | 58 | 44 | **91** |
| Microwave-1 | 46 | 22 | **86** |
| Average | 52 | 45 | **87** |

Table B.5: **Comparison of OPTIMUS vs. TAMP**. We plot percentage success on randomly chosen states from the environment. We find OPTIMUS greatly outperforms the TAMP supervisor, whether TAMP uses task space control or joint space.

evaluation on a machine with an RTX 3090 GPU and Intel i9-10980XE CPU and include the results in Table B.6. TAMP takes 150ms per action on average while OPTIMUS (30M parameters) takes 21ms per action and OPTIMUS (100M parameters) takes 31ms per action. TAMP pays a high up-front cost of 2-5 seconds, and then executes a feedback controller to quickly track the planned way-points. In contrast, OPTIMUS spends a constant amount of time per action. Furthermore, it is possible to greatly improve the inference time performance of OPTIMUS by employing techniques such as FlashAttention [78], model compilation, and TensorRT.

| TAMP | OPTIMUS (30M) | OPTIMUS (100M) |
|---|---|---|
| 150ms | 21ms | 31ms |

Table B.6: **Timing Results.** We measure the average time taken per action (lower is better). On average, OPTIMUS is 5-7.5x faster to execute than TAMP.

**By distilling TAMP, we obtain a performant policy that executes high-frequency *low-level* control from *purely perceptual* input.** OPTIMUS produces policies that are fast to execute, reactive and perform visuomotor control at similar performance to policies that have access to state information (Fig. C.2) and out-performs the privileged TAMP expert (Table B.5).

# C Ablations

In this section, we ablate components of OPTIMUS, low-level controller, data filtration scheme, gripper control scheme and data generation process, observation space design and loss function.

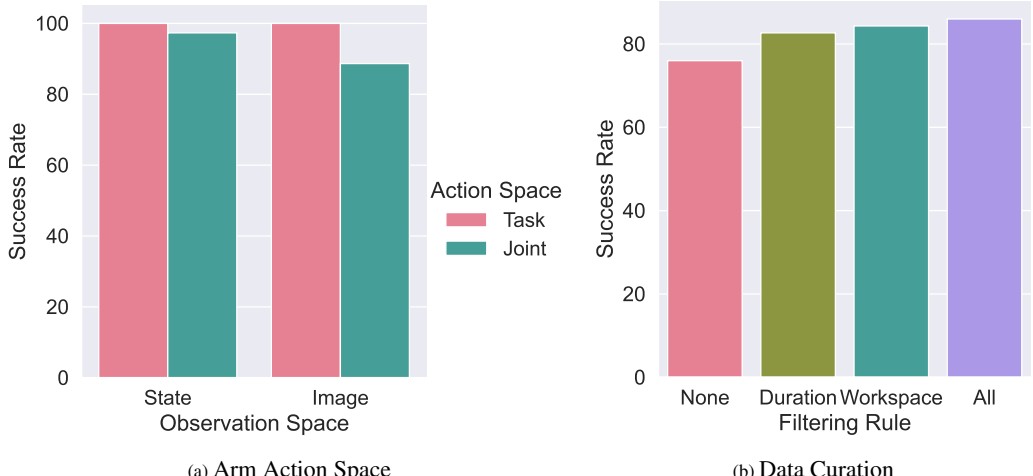

(a) Arm Action Space        (b) Data Curation

Figure C.1: **Effect of Arm Action Space Choice and Data Filtering Rules.** (a) OPTIMUS task success rate improves with task-space over joint-space actions when using image observations. Image observation-space policies perform comparably to the privileged state-based policies when using task-space actions. (b) Performance is improved by filtering TAMP success trajectories based on the visible workspace and their duration.

**Task-space control greatly improves visuomotor learning performance.** We evaluate different controllers on the Microwave-1 task. For state-based learning, we find that the choice of action space makes little difference; both control schemes achieve high performance (98% for joint space vs. 100% for task space). However, when training with visual observations, we find that there is a large gap (86% vs. 100%) in performance between joint control and task-space control. We hypothesize that this is due to the difficulty of learning an inverse kinematics mapping from visual input, *i.e.* mapping 2D pixel locations to 7DOF joint angles.

**Data filtration results in a significant improvement in policy success rates.** On the Microwave-1 task, we train four policies with different filtration schemes: 1) no filtering (None), 2) filtering based on trajectory length (Duration) 3) filtering based on visible workspace limits (Workspace), and 4) Duration and Workspace combined (Both). We find (Fig. C.1) that policies trained on unfiltered data perform worse when compared to those trained on filtered data. Specifically, workspace filtering has a greater impact than Duration. Combining both forms of filtering results in the greatest performance improvement of 10% and demonstrates that filtering TAMP trajectories is crucial to obtaining high success rates for learned policies.

**Discrete gripper control and short "stall" regions directly impact the performance of TAMP imitation.** We first analyze the impact of switching from continuous to discrete gripper control on the Stack task in Fig. C.2. By using discrete control, we can improve the success rate by 4%, while qualitatively we observe smoother gripper control and decisive grasps. On the other hand, we find that the decision to tune the length of "stall" regions, namely TAMP grasp and release actions, is crucial to the performance of OPTIMUS. As observed in Fig. C.2, reducing the number of control actions per grasp and release action greatly improves performance, from 78% at 25 steps to 100% at 5 actions. This is likely due to two reasons, 1) we shorten the overall length of the roll-outs, easing the learning burden, and 2) we reduce the likelihood of the policy to encounter a series of states where the observations and actions do not change, which can result in freezing behavior in the policy.

**Camera view selection enables greatly improved visuomotor learning.** We evaluate two camera views on the Stack task. Both camera poses keep all objects as well as the robot in view; one is

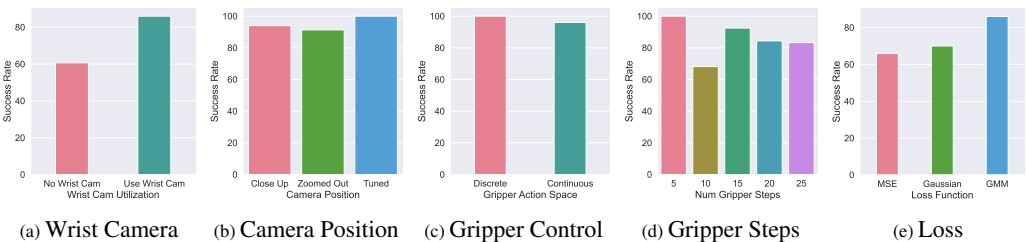

| (a) Wrist Camera | (b) Camera Position | (c) Gripper Control | (d) Gripper Steps | (e) Loss |

Figure C.2: **Effect of Observation, Action and Loss Decisions.** We ablate a variety of design decisions in OPTIMUS and demonstrate that each produces a clear improvement.

close up which hinders accurate estimation of scene geometry while the other is farther away which decreases the size of the objects in the frame, making it difficult for the policy to focus on them. As a result, we find in Fig. C.2 that a well-tuned camera view that is angled and positioned appropriately performs best. We additionally evaluate the impact of using a wrist camera. For tasks with primitive objects such as blocks, we found that the wrist cam had little impact. However, moving to tasks such as Microwave, where close up views of the handle and target object enable improved perception of grasp geometries, the wrist camera affords a significant performance improvement as we show in Fig. C.2.

**GMM loss enables OPTIMUS to better handle the multi-modality of TAMP supervision.** TAMP generates highly multi-modal action distributions through randomized planning and non-deterministic IK. Therefore, as we note in Sec. 3.3, we use Gaussian Mixture Models to model the multi-modality. We experimentally validate that GMM output distributions greatly improve learning performance by comparing against MSE loss, which produces a deterministic, uni-modal output distribution, and Gaussian log-likelihood, which produces a non-deterministic, uni-modal output distribution. We find that GMM loss greatly out-performs both output distributions (86% vs. 66% and 70%). While including a stochastic output distribution such as a Gaussian does improve performance by 4%, the multi-modality of GMM produces a further improvement of 16% performance. The results demonstrate that by providing the policy a more expressive output distribution, we can greatly improve how well the policy can model the TAMP expert.

## D  Environments

In this section, we provide a detailed description of the environments we use to evaluate OPTIMUS. We begin by describing settings which are common across environments. We then discuss each task individually.

For all tasks, we use a Franka Panda 7-DOF manipulator with the default Franka gripper, though the TAMP system is capable of generating supervision using any manipulator, provided the robot URDF. For the Stack task, we use the block stacking environment from Robosuite [58], modifying it to include up to 5 blocks and a larger workspace region. For all other tasks we use IsaacGym [79] with the PhysX [80] back-end. For each task, we use a fixed reset pose for the robot, while randomizing the positions of sampled objects. Object orientation about the z-axis is sampled uniformly at random from 0 to 360 degrees for all tasks.

For PickPlace, Multi-step PickPlace, Shelf and Microwave, we sample objects from ShapeNet [49]. We select objects that have valid grasps in the Acronym [51] dataset. We further refine our dataset by filtering out objects that do not simulate well in our IsaacGym environments. From the remaining objects, we form two datasets with 19 and 72 objects respectively.

We next provide additional details for each task.

**Stack**: The goal is to stack the blocks in a fixed ordering. Each block is a different color. The block positions are sampled uniformly in an area of size 28cm x 28cm. The base block is of size $2.5cm^3$; the rest are of size $2cm^3$. The task is considered solved if all of the blocks are stacked in the correct ordering.

**StackAdapt**: The task is the same as Stack, except there are two platforms, the blocks must be stacked on the target platform only. There is a 50/50 chance for the base block to be spawned on the target platform, in which the task simply involves stacking, and the base block to be spawned on the other platform, which requires the agent to first place the base block on the target platform then stack on top of it.

**PickPlace**: The task involves picking and placing ShapeNet objects from the left platform to the right platform. The platforms are of size .25 by .25 and are kept .5 apart. The object positions are sampled uniformly at random on the platform. The task success criteria is fulfilled if the object is placed anywhere on the target platform.

**Multi-step PickPlace**: The task involves picking and placing ShapeNet objects from platforms on the left to bins on the right. Up to four objects: a basket, vase, magnet or cup are sampled on separate platforms. Each platform is of size .15x.15 and each bin is of size .2x.2m. Each object's position is sampled uniformly at random on its associated platform. The task is solved when all objects are in their associated bins.

**Shelf**: The task involves moving ShapeNet objects from the lower rung of the shelf to the middle one. The shelf is 1m tall and has three rungs of size .5m x .25. The position and size of the shelf are constant. Object positions are sampled on the lowest rung, uniformly at random across the surface. The task is solved when the object is placed on the middle rung.

**ShelfReceptacle**: This task is the same as Shelf, but the shelf size is randomized within the following intervals: height (.8-1m), rungs: (.5x.25m - .4x.75m).

**Microwave**: The goal is to open the microwave by pulling open the handle, grasp a ShapeNet object, and place it inside the microwave. The microwave is .3m tall, 50cm wide and 20 cm deep. Microwave position and size are held fixed. The initial angle of the microwave door is 0, i.e. fully closed. Object positions are sampled on a platform of size .25x.25m. The agent has succeeded when the object is inside the microwave.

**MicrowaveReceptacle**: This task is the same as Microwave, but the microwave size is randomized within the following intervals: height (.3-.4m), width: (.5-.6m), depth: (.2-.3m).

**MicrowaveAdapt**: The task is the same as the microwave task, except with 50% probability an object is spawned in front of the microwave door, requiring the agent to first move the object aside then open the door and place the target object inside.

# E Agent Structure

**Observation spaces:** We use the same set of proprioceptive observations across all tasks: end-effector position, end-effector orientation (quaternion), gripper position. For each task, we select a different camera view that maximizes scene coverage. For Shelf and Microwave, we use two views, left and right shoulder views, whereas for the rest of the tasks we use a single forward facing view. Additionally, we use a wrist camera for every task, which greatly improves the performance. We use camera images of size 84x84. We empirically validate these decisions in Sec. C and visualize the results in Fig. C.2.

**Action spaces:** As mentioned in the main text, we use task space control for moving the arm. In Robosuite, we use the built-in OSC controller [59]. In IsaacGym, we used a simple IK-based task-space controller. With regard to gripper control, we discuss and resolve two challenges related to TAMP. 1) Continuous gripper actions produced by the TAMP solver can be challenging for the network to fit, as the network does not fully commit to predicting grasps. To that end, we modify the gripper actions to be binary open and close motions which improves performance and reduces noise in policy execution. We validate that this results in a performance improvement in Appendix C. 2) TAMP demonstrations can include"stall regions": segments of the trajectory in which the robot is not moving, such as when TAMP executes gripper-only actions for grasps and placements. This results in trained policies that may freeze after grasping an object, as the data does not contain cues for when to exit the stall region. To address this issue, we tune the length of stall regions during data collection against the agent's history length to ensure data collection success rate remains high while minimizing policy freezing behavior.

## F  Experiment Details

| Hyper-parameter | Value |
|---|---|
| Learning Rate | 0.0001 |
| Batch Size | 16/512 |
| Warmup Steps | 0 |
| Linear Scheduling Steps | 100K |
| Final Learning Rate | 0.00001 |
| Weight Decay | 0.01 |
| Gradient Clip Threshold | 1.0 |
| Number of Gradient Steps | 1M |
| Optimizer Type | AdamW |
| Loss Type | GMM |
| GMM Components | 5 |
| GMM Min. Std. Dev. | 0.0001 |
| GMM Std. Dev. Activation Fn. | SoftPlus |

Table F.1: Hyper-parameters used during training.

| | OPTIMUS (30M/100M) | MLP (30M/100M) | RNN (30M/100M) | BeT (30M/100M) |
|---|---|---|---|---|
| Num Layers | 6/12 | 2/6 | 2/3 | 6/12 |
| Hidden Dimension | | 1024/1024 | 1000/2000 | |
| Context Length | 8/8 | | 10/10 | 10/10 |
| Num Heads | 8/16 | | | 8/16 |
| Transformer Embed. Dim. | 256/512 | | | 256/512 |
| Embedding Dropout Prob. | 0.1/0.1 | | | 0.1/0.1 |
| Attention Dropout Prob. | 0.1/0.1 | | | 0.1/0.1 |
| Output Dropout Prob. | 0.1/0.1 | | | 0.1/0.1 |
| Positional Embed. | Learned/Learned | | | Learned/Learned |
| Positional Embed. Type | Relative/Relative | | | Relative/Relative |
| Num. Clusters | | | | 24/24 |
| Offset Loss Scale | | | | 100/100 |

Table F.2: Model hyper-parameters.

**Network and Training Details:** We include the model hyper-parameters for the 30M and 100M parameter variants of each method in Table F.2. For the vision-backbone, as discussed in the main text, we use a Resnet-18 [60] with a Spatial Softmax [61] output to encode each image separately. For details, please see the Robomimic paper [29]. We include learned positional embeddings with each token and employ relative, rather than absolute, position embeddings to enable the network to adapt to longer horizons at test time. We use a linear annealing schedule that reduces the learning rate from $10^{-4}$ to $10^{-5}$ over 100K gradient steps and then keeps the learning rate constant. We train with the AdamW optimizer with a weight decay of $0.01$ and no learning rate warm-up. For single-task learning, we train with a batch size of $16$ on a single V100 GPU, while for multi-task learning we train using batch size of $512$ to $1024$ depending on the task, across 8 V100 GPUs. For visuomotor learning, we train with multiple camera views with image size 84x84, and we augment the data with random crops [29, 81, 82]. We additionally list the hyper-parameters used for training in Table F.1. One note of interest: for multi-task training, we found that increasing the batch size greatly improved the results; hence we use a batch size of 512.

For BeT, we tried using the original authors codebase, which we augmented with our vision backbone, but found that the performance was extremely low. Instead, we re-implemented BeT as a modification of OPTIMUS, using the same network structure but predicting a discrete cluster center and offset head instead and training using the focal and MT losses from the BeT paper. We found that the standard hyper-parameters for BeT did not perform well, and after significant hyper-parameter tuning found that the combination of 24 cluster centers and offset loss scale of 100 performed best.

**Evaluation Protocol:** We note additional details regarding our evaluation protocol as follows. We split each dataset into a set of training and validation trajectories (using a 90/10 split). From the validation trajectories, we save the initial state of the demonstration. During evaluation, we reset the simulator state to an initial state from the validation set, and execute the policy from there. By comparing on the same set of validation states, we can better evaluate performance across seeds and algorithms. Note this means evaluation is performed from states that the TAMP solver is able to solve. As we note in Sec. 4.1, in practice this distinction matters little, as the TAMP system does not have a systematic failure case which could be passed on to the policy. Therefore we observe similar success rates when evaluating on randomly sample poses from the environment.

# G Related Work

## G.1 Offline Learning from Demonstrations

Imitation Learning (IL) is a paradigm for training robots to perform manipulation tasks by leveraging a set of expert demonstrations. In this work, we focus on offline learning, in which a policy learns a dataset of demonstrations, without any additional interaction. This is typically done through Behavior Cloning (BC) [30], in which a policy is trained to imitate the actions in the dataset through supervised learning. While this is a simple approach, it has proved incredibly effective for robotic manipulation [29, 31, 32, 33, 34, 35, 36, 37], particularly when coupled with a large number of demonstrations [10, 20, 38, 39]. Concurrent work has proposed leveraging Diffusion Models [83] to train policies via BC [84] in order to handle multi-modality of demonstrations. Our work instead focuses on how to best imitate TAMP with Transformers; Diffusion Policies, in particular their Transformer variants, could be straightforwardly integrated into OPTIMUS.

Human supervision is a common source of demonstrations. Several prior works use kinesthetic teaching [85, 86, 87, 88], in which a human manually guides an arm through a task, but this does not scale. Many works have leveraged teleoperation systems [13, 14, 35, 15, 89, 90, 91, 20, 38, 39], in which a human remote controls a robot arm to guide it through a task. However, scaling teleoperation is costly because it can require months of data collection and numerous human operators [10, 20, 89]. This has motivated the development of intervention-based systems, in which humans provide smaller corrective behaviors to an agent [92, 93, 94, 95, 96, 97], enabling more sample-efficient learning and less operator burden. Instead of relying on human operators for supervision, we learn policies from demonstrations provided by a TAMP supervisor, which can generate large, diverse datasets without human supervision.

## G.2 Transformers for Robot Control

Recent work explores the application of Transformers to controlling robot manipulators. Transformer-based policy architectures such as Gato [12], PerAct [40], VIMA [41], RT-1 [10], Dasari and Gupta [42], and Behavior Transformer [43] have demonstrated impressive results across a range of robotic manipulation tasks, yet make use of discretization of the input observations and output actions, limiting their applicability to tasks requiring precise manipulation. Additionally, PerAct [40] and VIMA [41] use abstracted actions to ease the learning burden at the cost of expressivity and execution speed. HiveFormer [71] is closest to our method in terms of architecture and training protocol but also assumes temporally-extended motion planner actions. As a result, these systems require privileged knowledge of the geometry of the environment to ensure safety. In contrast, OPTIMUS uses a Transformer architecture that is efficient to train and scale, fast-to-execute, consumes raw observations, and outputs low-level control actions.

## G.3 Task and Motion Planning

Task and Motion Planning (TAMP) [27] addresses controlling a hybrid system through planning a sequence of discrete of manipulation types (*task planning*) realized through continuous motions (*motion planning*). TAMP approaches consume kinematic or dynamic models [44] of individual manipulation types and search over combining them in a manner that achieves a goal. Classically, these models are engineered; however, recently, they have been learned using methods such as Gaussian Processes [64] or Deep Neural Networks [98, 65, 99]. These mixed engineering-learning TAMP techniques can be quite effective, but they impose a strong human design bias, capping policy performance. Also, they are too computationally expensive to be run in real-time, preventing them from quickly reacting to new observations.

There has been recent interest in approaches that imitate planning [45, 46, 47]; however, these approaches generally focus on single-step motion generation. The exception is [28], which recently proposed an approach, Guided TAMP, that directly imitates TAMP. Our work builds on this direction in several ways. First, Guided TAMP primarily addresses control from privileged state, while we

focus exclusively on visuomotor learning, which requires fewer assumptions. Second, Guided TAMP proposes a hierarchical policy that first predicts a discrete task-level action and then, conditioned on that action, predicts the next control. In order for the learner to predict a task-level action, they require a fixed set of ground actions, preventing the same policy from being deployed in tasks, for example, with varying numbers of objects. In contrast, our Transformer architecture does not explicitly reason about task-level actions and thus does not require grounding and fixing the objects in the scene. Finally, we identify new considerations when using TAMP as a data generation pipeline.

