# OpenReview forum: "Imitating Task and Motion Planning with Visuomotor Transformers"
_robot-learning.org/CoRL/2023/Workshop/TGR — CoRL 2023 Workshop TGR Poster_

### Official Review · Reviewer_sHcA · 2023-10-14

**Rating:** 8
**Confidence:** 3

**Review:**

This paper proposes a Task and Motion Planning module that can autonomously generate large-scale datasets of diverse demonstrations, and an imitation learning system to learn visuomotor Transformer policies. Both the demonstration generation and learning modules could be important steps towards generalist robots.

---

### Official Review · Reviewer_gG2q · 2023-10-20

**Rating:** 8
**Confidence:** 4

**Review:**

In this paper, a Task and Motion Planning module is proposed for the autonomous creation of large-scale datasets containing diverse demonstrations. I believe this work aligns well with the theme of acquiring versatile skills and scaling up robot learning.

---

### Decision · Program_Chairs · 2023-10-20

**Decision:**

Accept (Poster)

**Comment:**

Great paper and closely aligned topic!